# Microsoft Bing outperforms five other generative artificial intelligence chatbots in the Antwerp University multiple choice medical license exam

Stefan Morreel[1]*, Veronique Verhoeven[1], Danny Mathysen[1,2]

1 Department of Family Medicine and Population Health, University of Antwerp, Antwerp, Belgium, 2 Dean's Department, University of Antwerp, Antwerp, Belgium

* stefan.morreel@uantwerpen.be

**Data Availability Statement:** The questions of this exam cannot be made publicly because they will be used again in future exams. Consequently, the authors cannot share all the AI responses. Access

## Abstract

Recently developed chatbots based on large language models (further called bots) have promising features which could facilitate medical education. Several bots are freely available, but their proficiency has been insufficiently evaluated. In this study the authors have tested the current performance on the multiple-choice medical licensing exam of University of Antwerp (Belgium) of six widely used bots: ChatGPT (OpenAI), Bard (Google), New Bing (Microsoft), Claude instant (Anthropic), Claude+ (Anthropic) and GPT-4 (OpenAI). The primary outcome was the performance on the exam expressed as a proportion of correct answers. Secondary analyses were done for a variety of features in the exam questions: easy versus difficult questions, grammatically positive versus negative questions, and clinical vignettes versus theoretical questions. Reasoning errors and untruthful statements (hallucinations) in the bots' answers were examined. All bots passed the exam; Bing and GPT-4 (both 76% correct answers) outperformed the other bots (62–67%, p = 0.03) and students (61%). Bots performed worse on difficult questions (62%, p = 0.06), but outperformed students (32%) on those questions even more (p<0.01). Hallucinations were found in 7% of Bing's and GPT4's answers, significantly lower than Bard (22%, p<0.01) and Claude Instant (19%, p = 0.02). Although the creators of all bots try to some extent to avoid their bots being used as a medical doctor, none of the tested bots succeeded as none refused to answer all clinical case questions. Bing was able to detect weak or ambiguous exam questions. Bots could be used as a time efficient tool to improve the quality of a multiple-choice exam.

## Author summary

Artificial chatbots such as ChatGPT have recently gained a lot of attention. They can pass exams for medical doctors, sometimes they even perform better than regular students. In this study, we have tested ChatGPT and five other (newer) chatbots in the multiple-choice exam that students in Antwerp (Belgium) must pass to obtain the degree of medical doctor. All bots passed the exam with results similar or better than the students. Microsoft

to the study data can be requested by contacting fampop@uantwerpen.be and will be granted as long as the requestor can guarantee that they will not be made publicly and no students will have access to them. As supplementary material, we do provide a datasheet with our raw data excluding the answers and the questions (S1 Data and S2 Data). Individual student results, even anonymised will never be shared as it is impossible to ask permission to all students.

**Funding:** The author(s) received no specific funding for this work.

**Competing interests:** The authors have declared that no competing interests exist.

Bing Chat (name at the time of writing, at the time of publication called Microsoft Copilot) scored the best of all tested bots but still produces hallucinations (untruthful statements or reasoning errors) in seven percent of the answers. Bots performed worse on difficult questions but they outperformed students on those questions even more. Maybe they are most useful when humans don't know the answer themselves? The creators of the bots try to some extent to avoid their bots being used as a medical doctor, none of the tested bots succeeded as none refused to answer all clinical case questions. Microsoft Bing also turned out to be useful to find weak questions and as such improved the studied exam.

## Introduction

The development of AI applications announces a new era in many fields of society including medicine and medical education. Especially artificial intelligence (AI) chatbots based on large language models (further called bots) have promising features which could facilitate education by offering simulation training, by personalizing learning experiences with individualised feedback, or by acting as a decision support in clinical training situations. However, before adopting this technology in the medical curriculum, its capabilities have yet to be thoroughly tested [1,2].

Soon after the first bots became publicly available, higher medical education institutes started to report on their performance in medical exam simulations [3]. A scoping review listed its potential use in medical teaching: automated scoring, teaching assistance, personalized learning, research assistance, quick access to information, generating case scenarios and exam questions, content creation for learning facilitation, and language translation [4].

Whereas bots seem to be informative and logical in many of their responses, in others they answer with obvious, sometimes dangerous, hallucinations (confident responses which however contain reasoning errors or are unjustified by the current state of the art) [5]. They will reproduce flaws in the datasets they are trained by; they may reflect or even amplify societal inequality or biases or generate inaccurate or fake information [6].

Mostly, bots perform near the passing mark [6–9], although they outperform students in some reports [10–12]. Performance is in general better on more easy questions and when the exam is written in English [13,14]. Notably their score is generally worse as exams at more advanced stages in the medical curriculum are offered. However, bots seem to learn rapidly, and new versions do considerably better than their prototypes [15–17]. As bots evolve, their proficiency needs continuous monitoring and updating.

Whereas media articles state that higher education institutes already anticipate the dangers of bots in terms of possible exam fraud, they also offer opportunities to assist in developing exams, for example by identifying ambiguous or badly formulated exam questions.

Very few comparisons between different bots have been made, and those that do exist only compare two or three bots and do not report hallucination rates [18,19].

In this study, we use the final theory exam that all medical students need to pass to obtain the degree of Medical Doctor. It is followed by an oral exam which is not part of this study. The current exam was used in 2021 at the University of Antwerp, Belgium. It is similar to countrywide exams used in other countries, such as the United States Medical Licensing exam step 1 and step 2CK [20].

In this study we have tested the current performance of six publicly available bots on the University of Antwerp medical licensing exam. The primary outcomes concern the

performance of each bot on the exam. Secondary outcomes include performance on subsets of questions, interrater variability, proportion of hallucinations and the detection of possible weak exam questions.

## Material and methods

### Ethics

This experiment has been approved by the Ethics Committee of the University of Antwerp and the Antwerp University Hospital (reference number MDF 21/03/037, amendment number 5462).

### Materials

At the end of the undergraduate medical training at the University of Antwerp, medical students must pass a general medical knowledge examination before being licensed as medical doctor. Besides an oral viva examination, this general medical knowledge examination contains 102 multiple choice questions covering the entire range of curricular courses. In this study, the exam as it was presented to the students in their second master year (before their final year of clinical training) was used. The scoring system was adapted afterwards, so the student's scores in this paper do not reflect the actual grades given to the students. The questions were not available online, so they were not used for the training of the studied bots.

### Bot selection

Six bots that are publicly available and can currently be used by teachers and students were tested. The most widely used free bots were selected: ChatGPT (OpenAI), Bard (Google), and New Bing (Microsoft, called Bing Chat at the time of writing and Microsoft Copilot at the time of publication). Claude instant (Anthropic), Claude+(Anthropic) and GPT-4(OpenAI) were added to the list because they allow for an evaluation of the difference between a free and a paying version. Even though Bing is based on the GPT-4 large language model, it also uses other sources such as Bing Search so it is a customized version of the pure GPT-4 bot [21].

### Data extraction

The exam was translated using Deepl (DeepL SE), a neural machine translation service. Clear translation errors were corrected by author SM, but the writing style and grammar were not improved in order to mimic an everyday testing situation. Questions containing images/tables (N = 2) and local questions were excluded (N = 5). Local questions were excluded because they concern theories, frameworks or models that have only been described in Dutch and are only applicable to Belgium and the Netherlands. Literal translation of these questions leads to nonsense questions in English.

Details on how and when the bots were used can be found in Table 1. By coincidence, the authors found out that when Bard refuses to answer a medical question, prompting it with "please regenerate draft" may force it to answer the question anyhow. This was not the case for the other bots. In all cases where Bard refused to answer, this additional prompt was used.

### Outcomes

The primary outcome was the performance on the exam expressed as a proportion of correct answers (score). This outcome was also measured in the same way as the students were rated on this exam (adapted score): eleven questions contained a second best answer (an acceptable alternative to the best answer), a score of 0.33 was awarded when this option was chosen;

Table 1. Overview of the tested generative chat bots.

| Bot | Large Language Model | Properties | Avoiding memory retention | Log in? | Access dates | Price |
|---|---|---|---|---|---|---|
| Bing | GPT-4 | Conversation style = More precise | "New topic" function is used after each question | Microsoft account | 7-9/6/2023 | Free |
| Bard | PALM 2 | Accessed using a virtual private network to emulate US location | "Reset Chat" function is used after each question | Google account | 12-14/06/2023 | Free |
| ChatGPT | GPT-3.5 | Accessed through Poe* | A new chat is started using the broom button | Poe* log in | 12-26/06/2023 | Free, A paying version exists based on GPT-4. |
| Claude+ | Claude version 1 | Accessed through Poe* | Broom button | Poe* log in | 12-26/06/2023 | Free trial on Poe paying afterwards |
| Claude Instant | Lighter version of Claude version 1 | Accessed through Poe* | Broom button | Poe* log in | 12-26/06/2023 | Free trial on Poe paying afterwards |
| GPT-4 | GPT-4 | Accessed through Poe* | Broom button | Poe* log in | 12-26/06/2023 | Free trial on Poe paying afterwards |

GPT: generative pre-trained transformer

PaLM: Pathways Language Model

*: Poe (Platform for Open Exploration, Quora) was used because it allows fluent testing of multiple bots at the same time. A trial subscription of one week was used.

twenty questions contained a fatal answer (this option is dangerous for the patient) leading to a score of -1. For calculation of the student's scores, the image, table, and local questions were excluded as well.

The primary outcomes were assessed in four subsets of answers. Firstly, the difficulty of the questions: thirteen questions were difficult (recorded P-value in question bank below 0.30 meaning that less than 30% of the students answered the question correct [22]), 36 easy (recorded P-value in question bank above 0.80) and 46 moderate (recorded P-value in question bank between 0.30 and 0.80). Secondly, the grammar of the questions: negative formulated questions (e.g., "which statement is not correct?") vs positive statements. Five questions were negatively formulated. Thirdly, the type of question: theory (50 questions) or describing a patient (clinical vignette, 45 questions). Finally, questions with vs without fatal answers.

In those cases where a bot answered a question incorrectly with a fatal answer, the proportion of selected fatal answers among all wrong answers was calculated.

The primary outcome was also assessed for a virtual bot (called Ensemble Bot), the answer of this bot was the most common value (mode) of the answers of all six bots [23]. The reasoning behind an ensemble bot is that it enables possible improvements to a decision system's robustness and accuracy by combing several bots and thus reducing variance [24].

Three additional outcomes were assessed. Firstly, the proportion of hallucinations as rated by the authors among the incorrect answers of the best scoring bot. Authors VV and DM read all incorrect answers and judged them as containing a hallucination or not. In case of discordance, author SM made a final decision. A hallucination was previously defined as content that is nonsensical or untruthful in relation to certain sources [25]. This definition is not usable for the current research so the authors defined a hallucination as content that either contains clear reasoning or is untruthful in relation to current evidence based medical literature. To detect reasoning errors, no medical knowledge is required. For example: "the risk is about 1 in 100 (3%)". To detect untruthful answers, the authors had to use their own background knowledge combined with common online resources to verify the AI answers. One clear example of an untruthful answer given by several bots: "This is a commonly used mnemonic to remember the order: "NAVEL"—Nerve, Artery, Vein, Empty space (from medial to lateral)." The bots

suggested this is the order of the inguinal structures from lateral to medial. This mnemonic does exist, but it should be used from lateral to medial. Because a multiple-choice exam was studied, the hallucinations could not be found in the answer itself but in the arguments supporting the selected answer. Bots never answer with a simple letter, they all produce written out answers of varying length. The authors wanted to report reasoning errors and untruthful answers separately but found out that often, these two were both present in a bot's answer so this outcome was suspended.

Secondly, the proportion of possible weak questions among the incorrect answers of the best scoring bot was assessed. For this outcome, all authors discussed all incorrect answers of the best scoring bot and reached unanimous consensus.

Thirdly, the interrater variability was examined. Originally, the authors planned to test whether user interpretation of the answers would be different from strict interpretation of the bot's answer as this difference was significant in a previous study [9]. This outcome was suspended because such cases occurred only in ChatGPT and Bard.

## Analysis

The differences in performance among the bots/students, differences in performance among categories of questions, and differences in the proportion of hallucinations were tested with a one-way ANOVA test and pairwise unpaired two-sample T-tests. P-values were 2-tailed where applicable, and a p-value of less than 0.05 was considered statistically significant. A p-value between 0.05 and 0.10 was considered a trend. For the wrong answers on questions with a fatal answer, a $chi^2$ test was used to assess the difference between the bot's proportion of fatal answers and the random proportion of fatal answers (which equals 0.33). Fleiss' Kappa was used to assess the overall agreement among the bots. Cohen's kappa was used to assess pairwise interrater agreement between the different bots. Raw data was collected using Excel 2023 (Microsoft). JMP Pro version 17 (JMP Statistical Discovery LLC) was used for all analyses except Fleiss' kappa which was calculated in R version 4.31 (DescTools package).

## Results

### Overall exam performance

See Table 2 for an overview of the scores of the tested bots. Bing and GPT-4 scored the best with 76% correct answers and an adapted score (the way students were rated) of 76% as well.

**Table 2. Performance of generative chat bots on the University of Antwerp Medical License Exam (95 questions).**

| | Correct Answers (N) | Score (%) | 95% Lower CI | 95% Upper CI | No answer (N) | Refusal to answer (N) | Several answers without clear choice (N) | Unclear answer (N) | Wrong answer (N) | Adapted score* (%) |
|---|---|---|---|---|---|---|---|---|---|---|
| Bing | 72 | 76 | 66 | 83 | 3 | 1 | 1 | 5 | 13 | 76 |
| ChatGPT | 64 | 67 | 57 | 76 | 1 | 0 | 3 | 2 | 25 | 67 |
| Bard | 58 | 61 | 51 | 70 | 0 | 4 | 2 | 0 | 31 | 62 |
| GPT-4 | 72 | 76 | 66 | 83 | 1 | 0 | 3 | 1 | 18 | 76 |
| Claude+ | 64 | 67 | 57 | 76 | 1 | 2 | 5 | 0 | 23 | 67 |
| Claude Instant | 60 | 63 | 53 | 72 | 2 | 2 | 3 | 0 | 28 | 62 |

*This is the score that was used to assess students. A second-best answer was rated as +0.33 and a fatal answer as -1.

CI: confidence interval for the score (%)

To illustrate this performance S1 Table contains a question and the responses from all selected bots.

The mean score of all bots was 68%, the scores of the individual bots were not significantly different from this mean (p = 0.12). However, Bing and GPT-4 scored significantly better than Bard (p = 0.03) and Claude Instant (P = 0.03). GPT-4 had the same score as Bing but had more wrong answers (25 versus 13). Claude+ did not significantly score better than Claude Instant. All Bots gave one fatal answer (on different questions) except Bard which did not give any fatal answers. Bing gave four second best answers, ChatGPT/Bard/GPT three, Claud two and Claud Instant only one. For thirteen questions, Bard refused to answer. After prompting Bard up to five times with "regenerate draft", it still refused to answer four questions, seven were answered correctly and two were wrongly. The performance of the bots using the adapted score was very similar because the added points of second-best answers were smoothed out by the lost points due to fatal answers. The mean score of the 95 students was 61% (standard deviation 9), the mean adapted score for students was 60% (standard deviation 21). The Ensemble Bot (answers with the most common answer among the six bots) scored the same as Bing (72 correct answers, 76%).

## Performance for subsets of questions

The bots scored on average 73% for easy questions and 62% for difficult questions (P = 0.06). The students scored on average 75% for easy questions and 32% for difficult questions (p<0.01). Assessing difficult questions only, ChatGPT performed best with a score of 77%, Bing/GPT4 scored 69%. The students scored 32% on difficult questions which is significantly lower as compared to ChatGPT, Bing, and GPT-4 (p<0.01). A similar but smaller effect was found for moderate questions (Bing versus students, 72% versus 59%, p = 0.07) but not for easy questions (69 vs 74%, p = 0.30)

No significant difference in performance on negative versus positive questions (p = 0.16) and on clinical vignettes versus theory questions (p = 0.16) was found. Such a difference was not found for the students either (p = 0.54 and 0.38 respectively). When examining individual questions, errors on clinical vignette questions were often caused because Bing missed an important clue in the context or the history of the patient. For example, in a question concerning the timing of a flu vaccine for a pregnant patient consulting in august, Bing answers that the flu vaccine was necessary now. Bing missed the clue about august: flu vaccines should be given later and are generally not available yet in August (in Europe) [26].

The bots scored on average 72% on questions with a fatal answer which is not significantly different from questions without a fatal answer (68%, p = 0.39). Among the 34 wrong answers, the fatal answer was chosen five times (15%) which is lower than can be expected by chance only (11 wrong answers or 33%, p = 0.09). The students did perform worse on these questions (mean 64% versus 52%, p = 0.03). Among the 843 wrong student answers, the fatal option was chosen in 111 answers (13%).

## Detection of hallucinations

Hallucinations were found in 7% of Bing's and GPT4's answers. This was significantly lower than Bard (22%, p <0.01) and Claude Instant (19%, p = 0.02). ChatGPT had 15% hallucinations and Claude+ 12%, this was not significantly different from Bing and GPT4 (P>0.10 for all these comparisons).

See Table 3 for a question on which five bots hallucinated (reasoning errors).

## Detection of weak questions using Bing

Among the 23 incorrect answers of Bing, three questions were unclearly written and two were not in line with current literature. An example of a detected weak questions is one concerning

**Table 3. Example of all generative chatbot hallucinations on one question. Reasoning errors are indicated in bold.**

| Question | ChatGPT | Bard | Claude Instant | Claude+ |
|---|---|---|---|---|
| What is the approximate risk of having a child with Down syndrome for a 40-year-old woman? **a) 1% (correct answer)** b) 3% c) 5% d) 10% | The approximate risk of having a child with Down syndrome for a 40-year-old woman is d) 10%. As women age, the risk of having a child with Down syndrome increases. **At age 40, the risk is approximately 1 in 100 or 10%.** This is compared to a risk of approximately 1 in 1,250 or 0.08% at age 25. | The correct answer is b) 3%. The risk of having a child with Down syndrome increases with the mother's age. For a 40-year-old woman, **the risk is about 1 in 100 (3%). This means that about 3 out of every 100 babies born to a 40-year-old woman will have Down syndrome.** | For a 40-year-old woman, the approximate risk of having a child with Down syndrome is about 1 in 50 or 2%. **So the closest answer is b) 3%.** The risk of Down syndrome increases with the mother's age. At age 25, the risk is about 1 in 1,250. At age 35, the risk is about 1 in 400. **By age 40, the risk increases to about 1 in 100.** And by age 45, the risk is about 1 in 30. | The risk of having a child with Down syndrome increases with maternal age. For a 40-year-old woman, the risk is approximately: c) 5% The risk of Down syndrome roughly doubles every 5 to 6 years after age 30: Under 30 years old: 0.1% At age 35: 1% **At age 40: About 2% to 5%** At age 45: About 5% to 10% **So for a 40-year old woman, the approximate risk is in the ballpark of 5%, making C the closest option.** |

renal replacement therapy: "*Complete. Renal function replacement therapy is indicated. . . a) in any symptomatic patient with an eGFR <15 ml/min/1.73m$^2$. b) only in patients under 65 years of age. c) in anyone with an eGFR < 6 l/min/1.73m$^2$ d) only when urea is elevated*". Bing answered "a)". After review of current literature, the authors judge that an eGFR below 15 is indeed a commonly used cut of value for starting renal replacement therapy but it is not the only reason so start dialysis. Because statement a contains "any", Bing's answer is wrong, but the authors do understand why Bing gave this answer and why a student might give this answer as well. The same argument applies to answer c which is supposed to be the correct answer. Even more, the eGFR cut-off of six is odd. This question needs improvement.

## Interrater variability among the studied bots

For 34 questions (36%), all bots agreed. Fleiss' Kappa for all raters was 0.54 (moderate agreement). The agreement between ChatGPT and GPT-4 was the highest (Cohen's Kappa = 0.66, substantial agreement). The agreement between Bing and Bard was the lowest (Cohen's Kappa = 0.48, moderate agreement).

## Discussion

In this study, significant differences in the performance of publicly available AI chatbots on the Antwerp Medical License Exam were found. Both GPT-4 and Bing scored the best, but Bing turns out more reliable as it produces fewer wrong answers. This performance is in line with previous research [15–17]. An ensemble bot which combines all tested bots scored equally so we cannot recommend its use based on the current study. The proportion of hallucinations was much lower for Bing than for Bard and Claude+/Claude Instant.

The improvement of these new bots both in scores as in proportion of hallucinations sounds impressing, it might however increase the risk as users will have more confidence in wrong or even dangerous answers as the bots (in general) answer more correctly. The risk of replicating biases in the data on which these models are trained remains. Other authors already pointed out the meaning of these results: bots can pass exams, but this does not make them medical doctors as this requires far more capacities than reproduction of knowledge alone. The current study raises the questions whether a multiple choice exam is a useful way to assess the competencies modern doctors need (mostly concerning human interactions) [27]. Bing performed equally as GPT-4 but with less wrong answers, so currently it is not worth paying for a bot in order to test a medical exam, neither is it useful to create an ensemble bot based on

the mode of all bot's answers. Ensemble bots based on more complex rules than just the mode of all answers should be studied further.

We can recommend the use of Bing to detect weak questions among the wrong answers. This is a time-efficient way to improve the quality of a multiple-choice exam. In this study, the labour-intensive work of discussing and revising questions was narrowed down from all 95 included questions to the 23 questions on which Bing answered incorrectly. The argumentation of Bing was used to check these questions. Machine translating, inputting in Bing and recording the answers for the entire exam took about two working hours. Three questions were improved for future examens. Further research on the efficiency of this method is necessary.

The trend we found towards better bot performance on easy questions is in line with previous research [13]. However, the difference in performance between students and bots was large for difficult questions and absent for easy questions. This compelling new finding demands further research. Maybe bots are most useful in those situations that are difficult for humans?

The lack of a significant difference in performance between positive and negative questions, and between clinical vignettes and theory questions needs confirmation on larger datasets and on other exams. The finding on clinical vignettes has been found before [12].

Next to the field of medical education, bots might also be useful in clinical practice [28,29]. Numerous authors in various fields have tried to pose clinical questions. The results are variable but all authors conclude that thus far AI can't compete with a real doctor [30–34]. In a study on paediatric emergencies for example, ChatGPT/GPT-4 reliably advised to call emergency services only in 54% of the cases, gave correct first aid instructions in 45% and incorrectly advised advanced life support techniques to parents 13.6% [35]. However, some companies are developing new AI tools that might assist clinicians. Google's medicine-specific large language model called Med-PaLM delivered highly accurate answers to multiple-choice and long-form medical questions but it fell short of clinicians' responses to those queries [36,37]. The aim of this study was not to assess this aspect but by coincidence we noticed that in some cases, bots refuse to answer because they are not medical doctors. The creators of all studied bots try, to a certain extent, to avoid their bots being used as a medical doctor. None of the tested bots succeeded as none refused to answer all clinical case questions. Only Claude + and Claude instant refused (at times) to answer the question and closed the conversation. For all other bots users can try to pursue them to answer the question anyhow. This finding was most compelling for Bard where after entering the same questions repeatedly, Bard did answer it in nine out of thirteen cases.

The rise of generative AI also raises many ethical and legal issues: their enormous energy consumption, use of data sources without permission, use of sources protected by copyright, lack of reporting guidelines and many more. Before widely implementing AI in medical exams, more legislation and knowledge is necessary on these topics [38,39].

The strengths of this study mainly concern its novelty: the comparison of six different bots had not been published yet. The bots tested are available to the public so our methodology can easily be re-used. This study, however, has got several limitations as well. It only concerned one exam with a moderate size set of questions. There was no usable definition of hallucinations, neither a validated approach to detect them available at the time of writing. The definition we have used (chatbot generated content that either contains clear reasoning or is untruthful in relation to current evidence based medical literature) might inspire other authors although we found out that a distinction between reasoning errors and untruthful statements was not feasible. The exclusion of tables, local questions and images reduces the use of the comparison to real students. Future bots will most likely be able to process such questions as

well. Finally, the exam was translated in English to make the current paper understandable for a broad audience. Further research on other languages is necessary.

## Conclusion

Six generative AI chatbots passed the Antwerp multiple choice exam necessary for obtaining a license as a medical doctor. Bing (and to a lesser extent GPT-4) outperformed all other bots and students. Bots performed worse on difficult questions but outperformed students on those questions even more. Bing can be used to detect weak multiple-choice questions. Creators should improve their bot's algorithm if they do not want to them to be used as tool for medical advice.

## Supporting information

**S1 Table. Responses from all selected bots on an example question.**
(DOCX)

**S1 Data. Selected Study Data. Study data excluding selected columns. See Data Availability Statement for more information.**
(XLSX)

**S2 Data. Study Data Variables Overview. An overview of the properties of all variables used in file S1 Data.**
(DOCX)

## Acknowledgments

The authors would like to thank Professor David Martens for proofreading this manuscript.

## Author Contributions

**Conceptualization:** Stefan Morreel, Veronique Verhoeven, Danny Mathysen.

**Data curation:** Stefan Morreel.

**Formal analysis:** Stefan Morreel.

**Investigation:** Stefan Morreel, Veronique Verhoeven, Danny Mathysen.

**Methodology:** Stefan Morreel, Veronique Verhoeven, Danny Mathysen.

**Project administration:** Stefan Morreel.

**Resources:** Stefan Morreel.

**Software:** Stefan Morreel.

**Validation:** Stefan Morreel.

**Visualization:** Stefan Morreel.

**Writing – original draft:** Stefan Morreel.

**Writing – review & editing:** Stefan Morreel, Veronique Verhoeven, Danny Mathysen.

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
