## [Decision Letter · Decision Letter 0]

22 Nov 2023

PDIG-D-23-00311

Microsoft Bing outperforms five other generative artificial intelligence chatbots in the Antwerp University multiple choice medical license exam

PLOS Digital Health

Dear Dr. Morreel,

Thank you for submitting your manuscript to PLOS Digital Health. After careful consideration, we feel that it has merit but does not fully meet PLOS Digital Health's publication criteria as it currently stands. Therefore, we invite you to submit a revised version of the manuscript that addresses the points raised during the review process.

Please submit your revised manuscript within 30 days Dec 22 2023 11:59PM. If you will need more time than this to complete your revisions, please reply to this message or contact the journal office at digitalhealth@plos.org. Please include the following items when submitting your revised manuscript:

We look forward to receiving your revised manuscript.

Kind regards,

Imon Banerjee

Section Editor

PLOS Digital Health

Journal Requirements:

Additional Editor Comments (if provided):

Please address primary comments of the reviewers including primary contribution, proper literature review.

Reviewers' comments:

Reviewer's Responses to Questions

**Comments to the Author**

1. Does this manuscript meet PLOS Digital Health’s publication criteria? Is the manuscript technically sound, and do the data support the conclusions? The manuscript must describe methodologically and ethically rigorous research with conclusions that are appropriately drawn based on the data presented.

Reviewer #1: Yes

Reviewer #2: Yes

2. Has the statistical analysis been performed appropriately and rigorously?

Reviewer #1: Yes

Reviewer #2: Yes

3. Have the authors made all data underlying the findings in their manuscript fully available (please refer to the Data Availability Statement at the start of the manuscript PDF file)?

Reviewer #1: Yes

Reviewer #2: Yes

4. Is the manuscript presented in an intelligible fashion and written in standard English?

Reviewer #1: Yes

Reviewer #2: Yes

5. Review Comments to the Author

Reviewer #1: Thank you for the opportunity to review the manuscript titled "AI Chatbots in the Antwerp Medical License Exam." The paper endeavors to shed light on the performance capabilities of several publicly available AI chatbots in the realm of medical knowledge assessments. With the rapid growth and adoption of AI tools in various sectors, understanding their potential utility, strengths, and weaknesses in the field of medicine becomes imperative. Utilizing the University of Antwerp's Medical License Exam, the authors critically analyze the performance of six different chatbots, with particular attention to correct answers, hallucinations, and weaknesses in multiple-choice questions. The findings aim to provide clarity on the feasibility and potential roles of AI in medical examinations and its broader implications on medical education.

Here are some minor comments that will improve the manuscript:

Comment 1:

The ensemble bot's introduction in line 133 requires further elaboration. The citation provided broadly addresses ensemble methods. For a clearer comprehension of the study's scope, it would be beneficial to offer an overview of this ensemble bot's specifics, including how to use it, and its primary functionalities.Further, its role and comparative significance in relation to other bots merit some discussion.

Comment 2:

In line 265, the statement alluding to the creators' intentions to restrict chatbots from being utilized as medical doctors could use some evidential support. A citation or reference validating this assertion would enhance the credibility of the claim. Additionally, considering the advancements in AI medical applications, such as Google's MedPalm, it might be worthwhile to delve into the potential aspirations of some companies to position their chatbots as medical consultants.

Comment 3:

In line 196 I noticed an inconsistency in the reporting of p-values within the manuscript. Specifically, the use of a percentage sign after the p-value in the line "P=0.06%" is atypical. Generally, p-values are reported as decimals, such as "P=0.0006" or "P<0.01". I recommend standardizing the presentation of p-values throughout the manuscript for clarity and consistency. 

Comment 4:

The statement in the conclusion, "Bots should improve their algorithm if they do not want to be used as a medical," could be further refined for clarity, perhaps by rephrasing it to "used in a medical capacity."

Comment 5:

The suggestion that Bing can serve as a tool for identifying weaker questions is intriguing. Elaborating on the mechanics of this process and its efficiency relative to other methods would further solidify this proposition.

Reviewer #2: Good work. 1-Improve your references to the literature, have a literature review section. Inform your reader what other work did before and what it is your work is adding to the literature.

2- Organization: what the difference is between (overall) outcome and (detailed) results sections may be confusing to the reader.

6. PLOS authors have the option to publish the peer review history of their article (what does this mean?). If published, this will include your full peer review and any attached files.

**Do you want your identity to be public for this peer review?** For information about this choice, including consent withdrawal, please see our Privacy Policy.

Reviewer #1: No

Reviewer #2: Yes: Prof. Arya Rahgozar

---

## [Decision Letter · Decision Letter 1]

10 Jan 2024

Microsoft Bing outperforms five other generative artificial intelligence chatbots in the Antwerp University multiple choice medical license exam

PDIG-D-23-00311R1

Dear Dr. Morreel,

We are pleased to inform you that your manuscript 'Microsoft Bing outperforms five other generative artificial intelligence chatbots in the Antwerp University multiple choice medical license exam' has been provisionally accepted for publication in PLOS Digital Health.

Best regards,

Imon Banerjee

Section Editor

PLOS Digital Health

I would like to thank the author for addressing reviewer comments. Please correct the grammar and spelling errors before publications.

Reviewer Comments (if any, and for reference):

Reviewer's Responses to Questions

**Comments to the Author**

1. If the authors have adequately addressed your comments raised in a previous round of review and you feel that this manuscript is now acceptable for publication, you may indicate that here to bypass the “Comments to the Author” section, enter your conflict of interest statement in the “Confidential to Editor” section, and submit your "Accept" recommendation.

Reviewer #2: All comments have been addressed

2. Does this manuscript meet PLOS Digital Health’s publication criteria? Is the manuscript technically sound, and do the data support the conclusions? The manuscript must describe methodologically and ethically rigorous research with conclusions that are appropriately drawn based on the data presented.

Reviewer #2: Yes

3. Has the statistical analysis been performed appropriately and rigorously?

Reviewer #2: Yes

4. Have the authors made all data underlying the findings in their manuscript fully available (please refer to the Data Availability Statement at the start of the manuscript PDF file)?

Reviewer #2: No

5. Is the manuscript presented in an intelligible fashion and written in standard English?

Reviewer #2: Yes

6. Review Comments to the Author

Reviewer #2: Thank you for trying to address the feedback. There are some tiny English issues across, for example there is an extra "to" in line 313, remove. Use editor tools to detect and correct.

7. PLOS authors have the option to publish the peer review history of their article (what does this mean?). If published, this will include your full peer review and any attached files.

**Do you want your identity to be public for this peer review?** For information about this choice, including consent withdrawal, please see our Privacy Policy.

Reviewer #2: No
